# An Iterative Implementation of the Signal Space Separation Method for Magnetoencephalography Systems with Low Channel Counts

**DOI:** 10.3390/s23146537

**Published:** 2023-07-20

**Authors:** Niall Holmes, Richard Bowtell, Matthew J Brookes, Samu Taulu

**Affiliations:** 1Sir Peter Mansfield Imaging Centre, School of Physics and Astronomy, University of Nottingham, University Park, Nottingham NG7 2RD, UK; richard.bowtell@nottingham.ac.uk (R.B.); matthew.brookes@nottingham.ac.uk (M.J.B.); 2Cerca Magnetics Limited, Unit 2 Castlebridge Office Village, Kirtley Drive, Nottingham NG7 1LD, UK; 3Department of Physics, University of Washington, Seattle, WA 98195, USA; staulu@uw.edu; 4Institute for Learning and Brain Sciences, University of Washington, Seattle, WA 98195, USA

**Keywords:** optically pumped magnetometer, magnetoencephalography, SSS, MEG analysis

## Abstract

The signal space separation (SSS) method is routinely employed in the analysis of multichannel magnetic field recordings (such as magnetoencephalography (MEG) data). In the SSS method, signal vectors are posed as a multipole expansion of the magnetic field, allowing contributions from sources internal and external to a sensor array to be separated via computation of the pseudo-inverse of a matrix of the basis vectors. Although powerful, the standard implementation of the SSS method on MEG systems based on optically pumped magnetometers (OPMs) is unstable due to the approximate parity of the required number of dimensions of the SSS basis and the number of channels in the data. Here we exploit the hierarchical nature of the multipole expansion to perform a stable, iterative implementation of the SSS method. We describe the method and investigate its performance via a simulation study on a 192-channel OPM-MEG helmet. We assess performance for different levels of truncation of the SSS basis and a varying number of iterations. Results show that the iterative method provides stable performance, with a clear separation of internal and external sources.

## 1. Introduction

Magnetoencephalography [1,2] produces images of electrophysiological human brain activity with high spatial and temporal resolution via measurement of extracranial magnetic fields generated by neuronal currents in the brain. MEG is a powerful tool for neuroscience [3] with significant clinical applications, particularly in epilepsy [4], but presents a significant engineering challenge. The neuromagnetic fields are typically tens of femtotesla in strength, requiring highly sensitive magnetometers, such as superconducting quantum interference devices (SQUIDs), and operation inside a magnetically shielded room (MSR) to reduce interference from magnetic fields generated by sources external to the MSR (e.g., moving vehicles, elevators and mains electricity). 

However, the MSR will not completely attenuate all external signals, and some sources of interference may exist within the MSR itself (including other biomagnetic sources such as the heart and electronic devices such as cameras required for patient monitoring). These interference components can be many orders of magnitude larger than the signals of interest, obfuscating neuromagnetic data. The signal space separation (SSS) method [5,6] is routinely employed in MEG analysis and is a powerful tool for separating the underlying neuromagnetic signals of interest from raw data containing interference. Specifically, the SSS method employs a magnetostatic multipole expansion to distinguish the contributions of sources of magnetic field, which are internal and external to an array of magnetic field sensors which form the MEG helmet.

Recently, MEG systems based on optically pumped magnetometers (OPMs) have been developed (OPM-MEG, see Brookes et al. and Schofield et al. for reviews [7,8]). Unlike SQUIDs, OPMs can be flexibly placed [9,10] and mounted into lightweight helmets [11], allowing virtually unconstrained participant movement [12,13,14] and scanning across the lifespan [15,16]. OPMs can also be placed much closer to the scalp than SQUIDs (since they do not require cryogenics). The reduced source-to-sensor separation leads to theoretical gains in spatial resolution via an ability to sample higher spatial frequencies of the neuronal fields [17,18,19]. Although a key advantage of OPM-MEG, this increased field complexity requires a corresponding increase in the dimensions of the SSS expansion.

The performance of SSS relies on several parameters, including sensor calibration and accurate knowledge of the array geometry, but also the total number of channels available compared to the dimensions of the SSS basis [20]. At present, the total number of magnetic field measurements available from OPM-MEG systems lags the number of channels of their cryogenic counterparts (e.g., Rhodes et al. [21] reported a 174-channel OPM-MEG system, compared to ~300 channels in commercial cryogenic MEG systems). This reduced channel count, coupled with an increase in the complexity of the spatial topography of the neuromagnetic fields, ultimately leads to poor performance of SSS. 

Here, we describe an iterative approach to the SSS method, which exploits the hierarchical nature of the multipole expansion to enable stable implementation on OPM-MEG devices with a low channel count. We begin with an overview of SSS and an illustration of the issues encountered when applied directly to a simulated OPM-MEG sensor array with 192 channels. We then describe the iterative approach and conduct a simulation study to investigate its performance and the optimal number of iterations required, followed by a discussion of the results obtained.

## 2. Theory

### 2.1. The SSS Method

Briefly, the SSS method uses a multipole expansion to describe the range of magnetic fields which can be measured by an array of sensors. Assuming the region containing the sensors is free from sources of magnetic field, a scalar potential V(r) which obeys Laplace’s equation ($\nabla^{2}V=0$) can be expressed using a series expansion of solid harmonic functions as
(1)$$V\left(r\right)=\sum_{l=1}^{\infty }\sum_{m=-l}^{l}\alpha_{lm}\frac{Y_{lm}\left(\theta ,\varphi \right)}{r^{l+1}}+\sum_{l=1}^{\infty }\sum_{m=-l}^{l}\beta_{lm}{r^{l}Y}_{lm}\left(\theta ,\varphi \right)\;$$
where
(2)$$Y_{lm}\left(\theta ,\varphi \right)=\sqrt{\frac{2l+1}{4\pi }\frac{\left(l-m\right)!}{\left(l+m\right)!}}P_{lm}\left(\cos\left(\theta \right)\right)e^{im\varphi }\;\;\;\;\;$$
is the normalised spherical harmonic function *r*, θ and φ are spherical coordinates, $P_{lm}\left(\cos\left(\theta \right)\right)$ is the associated Legendre function and $\alpha_{lm}$, $\beta_{lm}$ are the weighting coefficients of each component. The spherical harmonic functions represent spatial oscillations that increase in spatial frequency with increasing values of l (and m). Thus, high expansion orders l correspond to complex patterns of the magnetic scalar potential and the magnetic field. The r dependence of each term determines the spatial characteristics of the fields described by each set. Those proportional to $r^{-(l+1)}$ are singular at the origin and those proportional to $r^{l}$ diverge at infinity. For a known MEG sensor array geometry, and an origin inside the array, the first series in Equation (1) then describes magnetic fields from sources inside the array and the second series describes sources external to the array. The signal vectors measured by the array corresponding to each term can be calculated and (by denoting the signal vectors of $\frac{Y_{lm}\left(\theta ,\varphi \right)}{r^{l+1}}$ as $a_{lm}$ and the signal vectors of ${r^{l}Y}_{lm}(\theta ,\varphi )$ as $b_{lm}$) any total measured signal vector can be expressed as
(3)$$\Phi =\sum_{l=1}^{\infty }\sum_{m=-l}^{l}\alpha_{lm}a_{lm}\;+\sum_{l=1}^{\infty }\sum_{m=-l}^{l}\beta_{lm}b_{lm}$$

By constructing two matrices $S_{in}$ and $S_{out}$ (containing the signal vectors $a_{lm}$ and $b_{lm}$) and vectors $x_{in}$ and $x_{out}$ (containing the vector weights $\alpha_{lm}$ and $\beta_{lm}$) the signal (i.e., Equation (3)) can be compactly expressed as
(4)$$\Phi =\left[S_{in}\;S_{out}\right]\left[\begin{array}{c} x_{in}\\ x_{out} \end{array}\right]=Sx.$$

To estimate the internal signal of interest (${\hat{\Phi }}_{in}$) from the measured signal Φ one can estimate a weights vector ($\hat{x}$) via the pseudo-inverse matrix $S^{+}$ as
(5)$$\hat{x}=\left[\begin{array}{c} {\hat{x}}_{in}\\ {\hat{x}}_{out} \end{array}\right]=S^{+}\Phi \;\;\;\;\;$$
followed by computing
(6)$${\hat{\Phi }}_{in}=S_{in}{\hat{x}}_{in}.\;\;\;\;$$

In practice, the series must be truncated, as the sensor array is not capable of characterising all possible spatial frequencies. In fact, the total number of basis vectors must be less than or equal to the number of channels ($N_{chans}$) in the array. This dimension is given as
(7)$$N_{dims}={\left(L_{in}+1\right)}^{2}+{\left(L_{out}+1\right)}^{2}-2\leq N_{chans}$$
where $L_{in}$ and $L_{out}$ are the truncation orders for the inner and outer subspaces respectively. Cryogenic MEG systems contain roughly 300 channels, meaning high truncation orders (e.g., typically $L_{in}=8$ and $L_{out}=3$ are used, giving $N_{dims}=95\ll N_{chans}$) are possible and the computation of $S^{+}$ is stable. 

### 2.2. SSS with OPM-MEG

Recently developed OPM-MEG systems feature far fewer channels (typically <64 sensors, but OPMs measure two or three components of magnetic field per sensor) and require higher truncation orders due to an increased proximity of sensors to the scalp. For example, Tierney et al. [19] suggested orders of $L_{in}=11$ and $L_{out}=5$, $N_{dims}=178$ would be needed to fully realise the potential of such systems. This reduction in $N_{chans}$, combined with an increase in $N_{dims}$ means computation of $S^{+}$ can be unstable.

To illustrate this point, we estimated the relative reconstruction noise of an array of 64 triaxial (192-channels) OPMs (cMEG Adult Large Helmet, Cerca Magnetics Limited, Nottingham UK) as shown in Figure 1. The relative reconstruction noise ($n_{r}$) is an estimate of the residual noise found by simulating random (spatially and temporally uncorrelated) noise across the sensor array. The noise signal vector ($\Phi_{noise}$) is used in Equation (6) to estimate the internal signal vector (${\hat{\Phi }}_{in,\;noise}$) and $n_{r}$ is calculated as
(8)$$n_{r}=\frac{\left\|{\hat{\Phi }}_{in,\;\;noise}\right\|}{\left\|\Phi_{noise}\right\|}\;\;\;\;\;$$
where $\left\|\Phi \right\|$ denotes the norm of the vector. The value of $n_{r}$ can be interpreted as an approximate factor by which the noise level of a signal vector will be amplified following the SSS operation, ideally it should be close to unity. We investigated the reconstruction noise by generating 100 random signal vectors (zero-mean Gaussian noise) and computing $n_{r}$ for values of $L_{in}=3-11$ and $L_{out}=3-5$ (the centre of mass of the sensor positions was the origin of the system) before averaging across the 100 repeats. All simulations were implemented in MATLAB (MathWorks Inc., Natick, MA, USA), and we used code from Tierney et al. [19] to calculate the S matrices. Figure 2 shows that both $n_{r}$ and the condition number of the matrix S exponentially increase with increasing complexity of the model, suggesting standard implementation of SSS is likely to be unstable for this set-up. 

### 2.3. An Iterative Approach 

To address this and enable the exploration of SSS on OPM-MEG data, we implemented an iterative method for estimating the weights. We exploit the assumption that the SSS vectors represent MEG data in a hierarchical manner, i.e., we assume that lower-order components always explain a larger amount of signal energy than higher-order components (distal interference signals, with simple spatial field patterns, are high amplitude compared to the low amplitude and focal neuromagnetic signals, they will therefore dominate the MEG data). We first compute a subset of weights for a subset of the vectors of S, which initially includes only the first-order inner terms and all outer terms. We then create a new subset, this time including only the second-order inner terms and all outer terms, subtract our first estimate of the inner signal from the measured signal, and compute the subset of weights for the second-order terms (such that we only update the specific multipole components described by our subset of S) and update weights for this subset. This process repeats for all orders of the inner subspace and then iterates multiple times until a stable weights vector estimate is found.

First, we separate the (column-normalised) inner subspace vectors corresponding to orders $l_{in}=1\;to\;L_{in}$ as
(9)$$S_{in}=\left[S_{in,\;\;l_{in}=1}\;S_{in,\;l_{in}=2}\ldots \;S_{in,\;{l_{in}=L}_{in}}\right],\;\;\;\;\;$$
and then extract each set of vectors $S_{in,\;l_{in}}$ in turn to compute a series of $L_{in}$ partial bases, all including the outer subspace, as
(10)$$S_{l_{in}}=\left[S_{in,\;\;l_{in}}\;S_{out\;}\right].\;\;\;$$

We then apply the same approach to the weights vector
(11)$$x_{in}={\left[x_{in,\;\;l_{in}=1}\;x_{in,\;l_{in}=2}\ldots \;x_{in,\;{l_{in}=L}_{in}}\right]}^{T},\;\;\;\;\;$$
where T denotes the transpose, and create a series of $L_{in}$ partial weights
(12)$$x_{l_{in}}={\left[x_{in,\;\;l_{in}}\;x_{out\;}\right]}^{T}.\;\;\;\;\;$$

Starting with zero values for all weights, $l_{in}=1$, and a measured signal vector Φ, we estimate $x_{l_{in}=1}$ as
(13)$$x_{l_{in}=1}=S_{l_{in}=1}^{+}\Phi ,$$
and update the corresponding components of x. We then move to $l_{in}=2$, first subtracting the $l_{in}=1$ estimate from the measured signal and computing the $x_{l_{in}=2}$ specific weights as
(14)$$x_{l_{in}=2}=S_{l_{in}=2}^{+}\left(\Phi -Sx\right)=S_{l_{in}=2}^{+}\left(\Phi -S\left[\begin{array}{c} x_{in,l_{in}=1}\\ 0\\ \begin{array}{c} \vdots \\ 0 \end{array} \end{array}\right]\right).\;\;\;\;\;$$

For $l_{in}=3$ we evaluate
(15)$$x_{l_{in}=2}=S_{l_{in}=2}^{+}\left(\Phi -S\left[\begin{array}{c} x_{in,l_{in}=1}\\ x_{in,l_{in}=2}\\ \begin{array}{c} 0\\ \vdots \\ 0 \end{array} \end{array}\right]\right).\;\;\;\;\;$$

The process is repeated up to $l_{in}=L_{in}$, and then the entire cycle is iterated $N_{it}$ times according to
(16)$$x_{l_{in}}=S_{l_{in}}^{+}\left(\Phi -Sx_{\left\{l_{n}\in N|l_{n}<L_{in}\;and\;l_{n}\neq l_{in}\right\}}\right),$$
where $x_{\left\{l_{n}\in N|l_{n}<L_{in}\;and\;l_{n}\neq l_{in}\right\}}$ denotes only the weights corresponding to all orders except the specific value of $l_{in}$; these weights are already zero if the current iteration number ($n_{it}$ of $N_{it}$ total iterations) is 1, but for $n_{it}>1$, the non-zero weights calculated in the previous iteration must be replaced by zeros. The $l_{out}$ weights are always zero and update on each iteration.

Once the process is completed, a final weights vector ${\hat{x}}_{it}={\left[{\hat{x}}_{it,in}\;{\hat{x}}_{it,out}\right]}^{T}$ is formed, and the inner signal can be estimated as
(17)$${\hat{\Phi }}_{in}=S_{in}{\hat{x}}_{it,in}.\;\;\;\;\;$$

To assess the performance of the iterative approach, we returned to the reconstruction noise simulation. We used the same array geometry, signal vectors and range of $L_{in}$/$L_{out}$ values to compute $n_{r}$. We performed $N_{it}=10$ iterations for each signal vector. Compared to the pseudoinverse method, Figure 2 shows that the iterative method results in a marked decrease in reconstruction noise which was <2 for all cases c.f. ≫10, previously, indicating a more stable implementation of SSS is achieved. Our MATLAB implementation of the iterative method is provided as Appendix A.

## 3. Simulation Study

### 3.1. Methods

#### 3.1.1. Reconstruction Noise

We used the same simulated signal vectors from Section 2.2, but this time computed the relative reconstruction noise estimate following each iteration.

#### 3.1.2. Source Separation 

We simulated a series of magnetic dipoles placed inside and outside the sensor array. External dipoles were randomly positioned and oriented within a spherical shell with an inner radius of 2 m and outer radius of 3 m. Internal dipoles were randomly positioned and oriented on a spherical shell with an inner radius of 0.005 m and outer radius of 0.05 m; the minimum distance between the internal sources and the sensors was 21.5 mm. Both shells were centred at the centre of mass of the OPM-MEG helmet, as shown in Figure 3. We randomly chose 5 internal dipoles and 5 external dipoles and simulated signal vectors following the application of sinusoidal currents at distinct frequencies (randomly chosen integers between 1 and 100 Hz) for 1 second at a sample rate of 1200 Hz. The internal dipoles had a dipole moment of 10 nAm, and the external dipoles had a dipole moment of 10 mAm. We added zero-mean Gaussian noise of amplitude 30 fT to each simulated sensor. Signal vectors were calculated for each dipole in turn and summed to obtain the final vector. We then applied the iterative SSS method to the data. An example of the signals and the impact of standard implementation of the SSS method is shown in Figure 4. 

To assess the impact of the number of iterations, we applied the iterative SSS method using $N_{it}=20$ and extracted three metrics for each iteration: (1) the explained variance (EV) of the reconstructed inner signal compared to the calculated inner signal; (2) the root mean square error (RMSE) between the reconstructed inner signal and the known simulated inner signal; (3) the norm of the difference between the weights vectors for the current and previous iteration, i.e., $∆x_{it}=\left\|x_{n_{it}}-x_{n_{it}-1}\right\|$ for $n_{it}>1$. This was repeated for 100 different combinations of dipoles, and each signal vector was evaluated for $L_{in}=7-11$ and $L_{out}=3-5$. Results were then averaged over the 100 runs. 

### 3.2. Results

Figure 5 shows that an initial minimum in the relative reconstruction noise is found with the value then gradually increasing as the number of iterations increases. The noise level also increases as $L_{in}$ increases. The number of iterations after which the minimum point occurs decreases with an increase in $L_{out}$. In all cases, the value of $n_{r}$ is relatively low, roughly between 0.9 and 2.

Figure 6 shows that the SSS reconstruction of the internal signals explains >99% of the variance in the simulated signals for $L_{in}>9$ after five iterations. This increases to 99.8% explained variance for $L_{in}=11$ and $L_{out}=3\;or\;4$ reducing to 99.6% for $L_{out}=5$. The high levels of explained variance indicate the iterative method can accurately separate fields from internal and external sources.

Figure 7 reflects the results of Figure 4, showing a decrease in RMSE, which plateaus after five iterations. The final error value decreases further with an increase in $L_{in}$. Again, indicating good performance of the iterative approach.

Figure 8 shows the change in weights estimate $∆x_{it}$ decreases to <0.1 after five iterations. The change in weights is broadly consistent for all values of $L_{in}$, suggesting stable solutions are possible even for high-order models. 

## 4. Discussion

Application of the iterative implementation of the SSS method produces stable solutions with low relative reconstruction noise, high explained variance and low RMSE with a stable weights vector estimate ($∆x_{it}<0.1$) after just five iterations. We note that the results found may be specific to the specific array considered here. A key advantage of OPM-MEG is that, unlike cryogenic-based systems, OPM arrays are easily reconfigurable. An analysis like that presented above will be essential to assess the expected performance of the iterative approach on a case-by-case basis. For example, we note that the spherical shell used here to simulate the internal dipole signals (Figure 3a,b) does not represent the volume of a typical brain within the array and that the incorporation of more realistic internal sources may be useful before application to real data. By simulating relevant performance metrics, one can assess an appropriate number of iterations to use for optimum performance. When applied to real data (in which case the RMSE and explained variance cannot be estimated), a stopping condition based on the change in the weights estimate could be implemented by monitoring $∆x_{it}$ for each iteration and stopping when a certain value is found.

Although our iterative approach was developed to overcome issues associated with low channel counts, application of the SSS method to the rapidly developing field of on-scalp MEG (including high-Tc SQUIDs [22] as well as OPMs) and other biomagnetic measurements poses many opportunities for research. The optimisation and practical testing of array design (no longer limited by the confines of a cryogenic dewar), exploitation of triaxial sensing elements for the maximal separation of the internal and external subspaces (the average angle between the inner and outer subspaces for the 192-channel system studied is ~60° compared to ~10° for a commercial 306-channel cryogenic system [23], an increased subspace angle leads to improved shielding effects) and the introduction of a moving sensor array are all key areas to investigate. As it is a spatial method, SSS requires precise sensor calibration and accurate knowledge of the array geometry (to ensure the SSS vectors can capture all parts of the measured data). In practice, an SSS-guided calibration is often used to ensure optimal performance, but the operational principle of OPMs, and the potential for many different array configurations means SSS-guided approaches may be challenging to implement (as sensor calibration may vary depending on parameters, including the density of atoms in the OPM vapour cell and the background field experienced by the sensor). Methods based on the use of electromagnetic coils have been proposed in mitigation [24]. All these issues will need to be addressed to fully realise the potential of OPM-MEG.

## 5. Conclusions

Application of our iterative implementation of the SSS method to OPM-MEG systems with low channel count substantially reduces the relative reconstruction noise compared to the standard implementation. For our study, using a simulated 192-channel system, we found that a stable estimate was obtained after just five iterations, with little dependency on model complexity. Further study is needed to investigate the many array geometries afforded by OPM-MEG and apply the method to experimental data.

## Figures and Tables

**Figure 1 sensors-23-06537-f001:**
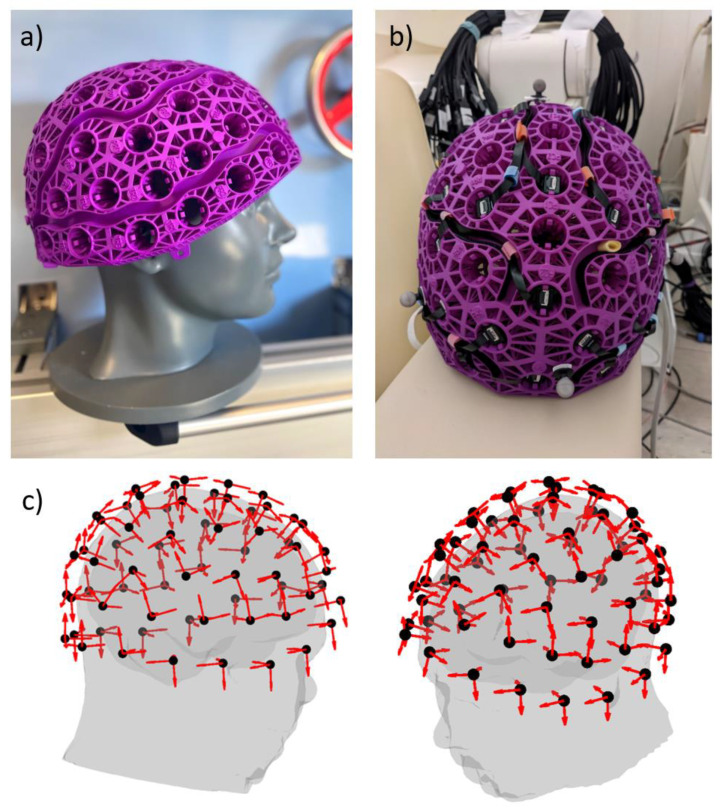
(**a**) The 64-slot Cerca cMEG helmet used in this study. (**b**) Incorporating 64 triaxial OPMs (QuSpin Inc., Louisville, CO, USA) gives a sensor array featuring 192 channels. (**c**) The position (black circles) and orientation (red arrows) of the channels are shown relative to the head and brain of a participant.

**Figure 2 sensors-23-06537-f002:**
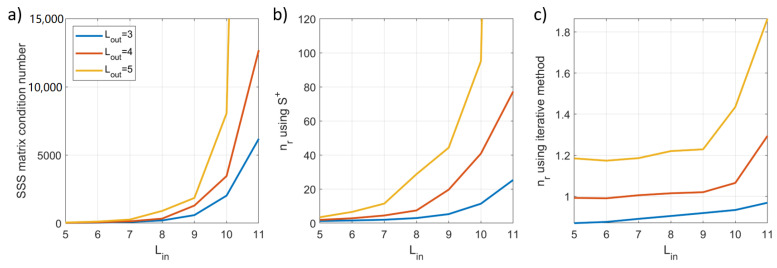
The impact of conventional implementation of SSS on the 192-channel array and the effects of an iterative method to stabilise the outputs. (**a**) The condition number of the (column-normalised) SSS matrix increases exponentially as a function of increasing complexity of the inner and outer subspaces. (**b**) An increasing condition number leads to an unstable computation of the pseudoinverse matrix, which causes an exponential increase in the reconstruction noise. (**c**) Application of an iterative approach to SSS significantly reduces the reconstruction noise estimate (note the difference in scales between (**b**,**c**)).

**Figure 3 sensors-23-06537-f003:**
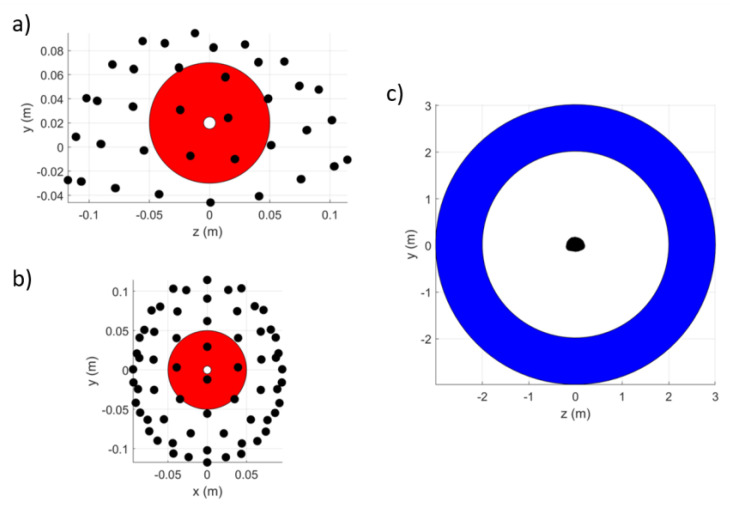
Spherical shells within which internal and external sources were generated. (**a**) A side-on view of the OPM-MEG helmet showing sensor locations (black dots) and the (red area) spherical shell (0.005 m < radius < 0.05 m) within which the internal dipole sources were generated. (**b**) A top-down view of the internal source shell. (**c**) A side-on view of the OPM-MEG helmet relative to the (blue area) spherical shell (2 m < radius < 3 m) within which the external dipole sources were generated. Both shells were centred at the centre of mass of the sensor locations: (x,y,z)=(0,0,0.02) m.

**Figure 4 sensors-23-06537-f004:**
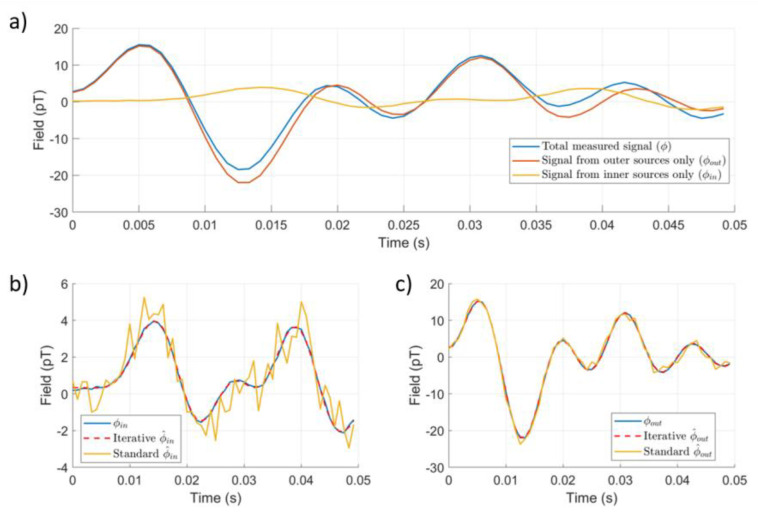
Performance of the iterative implementation of the SSS method for an example instance of simulated data. (**a**) A total of 50 ms of simulated data for a single channel in the OPM-MEG helmet. The total signal (Φ) is shown, as well as the signal from the internal sources ($\Phi_{in}$ ) generated from the five randomly positioned and oriented internal dipoles and the signal from the external signal ($\Phi_{out}$ ) generated by the five random external dipoles. (**b**) The simulated internal signal is compared to the SSS estimated inner signals (${\hat{\Phi }}_{in}$ ) found using both our iterative method and using a standard implementation of the SSS method ($L_{in}=10$ and $L_{out}=4$ for both cases, five iterations). (**c**) Similar plots for the external signal. We note that for both inner and outer signals, the iterative reconstructions agree well with the simulated data, but the standard implementation results in a noisy signal.

**Figure 5 sensors-23-06537-f005:**
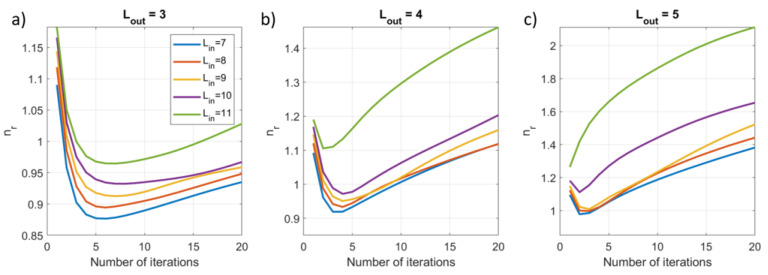
The impact of the number of iterations on the relative reconstruction noise for increasing model complexity, $L_{in}=7-11$ and (**a**) $L_{out}=3$, (**b**) $L_{out}=4$, (**c**) $L_{out}=5$, when using the iterative SSS approach.

**Figure 6 sensors-23-06537-f006:**
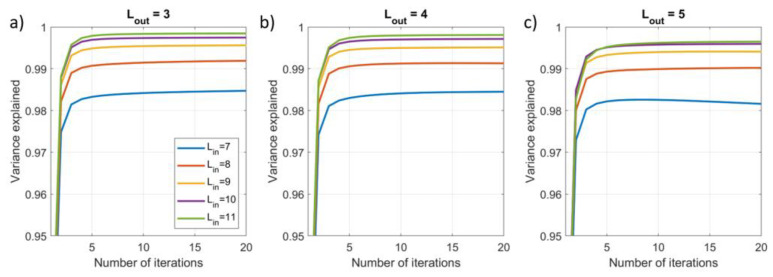
The impact of the number of iterations on the variance of the (simulated) inner signal that is explained by the array for increasing model complexity, $L_{in}=7-11$ and (**a**) $L_{out}=3$, (**b**) $L_{out}=4$, (**c**) $L_{out}=5$, using the iterative SSS approach.

**Figure 7 sensors-23-06537-f007:**
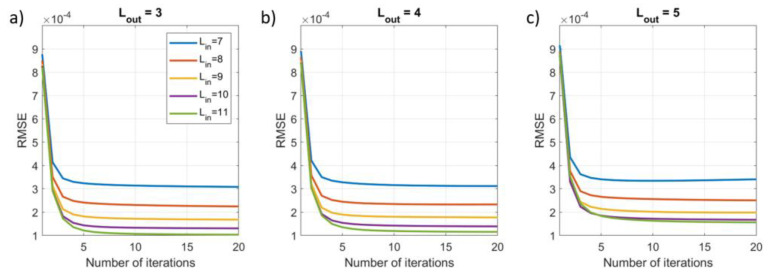
The impact of the number of iterations on the root mean square error of the simulated inner signal as compared to the reconstructed inner signal for increasing model complexity, $L_{in}=7-11$ and (**a**) $L_{out}=3$, (**b**) $L_{out}=4$, (**c**) $L_{out}=5$.

**Figure 8 sensors-23-06537-f008:**
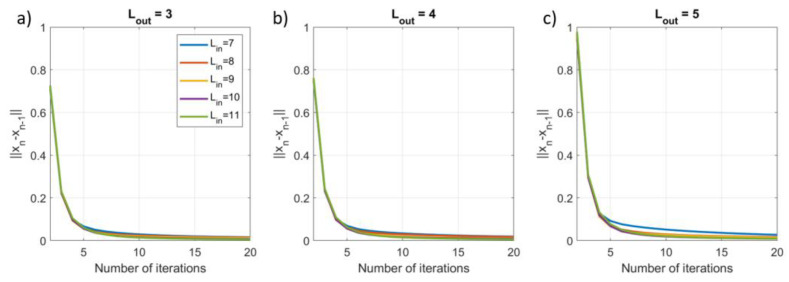
The impact of the number of iterations on the norm of the difference between the current estimate of the weights vector and the previous estimate for increasing model complexity, $L_{in}=7-11$ and (**a**) $L_{out}=3$, (**b**) $L_{out}=4$, (**c**) $L_{out}=5$.

## Data Availability

No data were acquired for this study. We acknowledge freely available code from Tim M. Tierney’s GitHub page, https://github.com/tierneytim/OPM (accessed on 20 November 2022) which was used to compute the SSS basis vectors.

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
