# Peer review of "An Iterative Implementation of the Signal Space Separation Method for Magnetoencephalography Systems with Low Channel Counts"

_sensors, 2023, doi:10.3390/s23146537_

Round 1

Reviewer 1 Report

Below I indicate my comments by section:

SSS section with OPM-MEG

- It is important to define the noise due to which it is caused, what kind of artifacts are present and if they can be reduced by what percentage.

Simulation Study Session

- A description of SSS reconstruction of iterative noise is made. 

Discussion

- Improvements to conventional analysis systems are proposed with a reduction in a significant number of channels.

- The analysis of the results by the difference of weight vectors is not deep.

- A more in-depth description of the problems that may arise when implementing this type of analysis is lacking.

- Further work to be done in the future

The writing style should be revised, especially in the introduction and discussion sections. 

Reviewer 2 Report

The paper is very interesting, the mathematical part is described very well.

The authors' own conclusions are not clear, it seems to me that an emphasis should be placed on the method magnetoencephaloaphy.

English is adequate and technical and research English is also adequate.

Reviewer 3 Report

Dear authors, thank you for submitting your manuscript to Sensors. Even though you only report on simulations, the findings are of interest to your community. I have two main suggestions:

- In the introduction, could you formulate a research question? I think it is along the line of "What is the optimal number of iterations required for iterative SSS, and how much better is it than conventional SSS"

- It will help the readers however if you do not only present your data, but also draw conclusions. I believe it is along the line of:

* Application of iterative SSS reduces the reconstruction noise by two orders of magnitude.

* The optimum number of iterations is dependent on the model complexity, and decreases with model complexity. Optimum values are from three to six.

* The proposed iterative SSS reconstruction can accurately separate fields from internal and external sources, with an explained variance of more than 99%. 

* The root mean square error between the reconstructed inner signal and the known simulated inner signal decrease, as well as the change in weights of the model, decrease rapidly with the number of iteration until five iterations.

Than minor remarks:

- Line 108, maybe remind the reader that higher truncation is needed because the sensors are close to the scalp

- Line 138, explain why you can assume low order components carry more energy

- Figure 2, also start the last graph from y=0 for fair comparison

Round 2

Reviewer 3 Report

You did not really understand my suggestions. So be it.

Author Response

We are sorry to hear that we did not respond to your comments in a satisfactory manner.

We appreciated the suggestion to include a research question in the Introduction and Conclusions of our manuscript, whilst a distinct question was not added, we did expand the Introduction to thoroughly describe the work presented in our paper and added a Conclusions section containing the relevant information suggested.

We also added explanations where prompted and decided against updating the plotting ranges due to a large difference in scales.